# Inhaled Ivermectin-Loaded Lipid Polymer Hybrid Nanoparticles: Development and Characterization

**DOI:** 10.3390/pharmaceutics16081061

**Published:** 2024-08-12

**Authors:** Seyedeh Negin Kassaee, Godwin A. Ayoko, Derek Richard, Tony Wang, Nazrul Islam

**Affiliations:** 1Pharmacy Discipline, School of Clinical Sciences, Faculty of Health, Queensland University of Technology (QUT), Brisbane, QLD 4000, Australia; 2School of Chemistry and Physics, Science and Engineering Faculty, Queensland University of Technology (QUT), Brisbane, QLD 4000, Australia; 3Centre for Genomics and Personalised Health, School of Biomedical Sciences, Queensland University of Technology (QUT), Brisbane, QLD 4000, Australia; 4Central Analytical Research Facility, Institution for Future Environment, Queensland University of Technology (QUT), Brisbane, QLD 4000, Australia; 5Centre for Materials Science, Queensland University of Technology, Brisbane, QLD 4000, Australia; 6Centre for Immunology and Infection Control (CIIC), Queensland University of Technology (QUT), Brisbane, QLD 4000, Australia

**Keywords:** Ivermectin, lipid polymer hybrid nanoparticles, pulmonary drug delivery, dry powder inhaler, lung cancer

## Abstract

Ivermectin (IVM), a drug originally used for treating parasitic infections, is being explored for its potential applications in cancer therapy. Despite the promising anti-cancer effects of IVM, its low water solubility limits its bioavailability and, consequently, its biological efficacy as an oral formulation. To overcome this challenge, our research focused on developing IVM-loaded lipid polymer hybrid nanoparticles (LPHNPs) designed for potential pulmonary administration. IVM-loaded LPHNPs were developed using the emulsion solvent evaporation method and characterized in terms of particle size, morphology, entrapment efficiency, and release pattern. Solid phase characterization was investigated by Fourier transform infrared spectroscopy (FTIR), differential scanning calorimetry (DSC), and thermogravimetric analysis (TGA). Using a Twin stage impinger (TSI) attached to a device, aerosolization properties of the developed LPHNPs were studied at a flow rate of 60 L/min, and IVM was determined by a validated HPLC method. IVM-loaded LPHNPs demonstrated spherical-shaped particles between 302 and 350 nm. Developed formulations showed an entrapment efficiency between 68 and 80% and a sustained 50 to 60% IVM release pattern within 96 h. Carr’s index (CI), Hausner ratio (HR), and angle of repose (θ) indicated proper flowability of the fabricated LPHNPs. The in vitro aerosolization analysis revealed fine particle fractions (FPFs) ranging from 18.53% to 24.77%. This in vitro study demonstrates the potential of IVM-loaded LPHNPs as a delivery vehicle through the pulmonary route.

## 1. Introduction

Pulmonary drug delivery is increasingly gaining attention as a desirable and non-invasive method for treating various medical conditions, particularly those affecting the lungs [1]. The pulmonary route exhibits superiority over both oral and parenteral routes, as the lungs provide a large surface area for drug absorption, avoid first-pass metabolism, and provide high local concentration and quick onset of action [2,3]. Inhaled drug delivery helps deposit drugs directly in the target tissue, resulting in comparable therapeutic efficiency with a lower amount of drug compared to systemic administration. This reduces exposure of drugs to non-target sites and, consequently, reduces undesirable side effects [4].

Various critical pulmonary diseases such as lung cancer, asthma, cystic fibrosis (CF), chronic obstructive pulmonary disease (COPD), and severe infections, including COVID-19, can impact the lungs, often resulting in significant mortality rates [5]. Lung cancer (LC) is one of the most commonly diagnosed cancers and ranks as the top contributor to cancer-related fatalities globally, accounting for approximately 2 million new cases and 1.8 million deaths annually [6,7]. Non-small cell lung cancer (NSCLC) accounts for approximately 85% of lung cancer cases and is often associated with poor prognosis owing to its tendency for late detection, high incidence of metastasis, and an increased likelihood of relapse [8]. Therefore, surgical intervention is not feasible at these late stages, which makes chemotherapy the primary treatment option [9]. Despite the pivotal role of chemotherapy in the treatment of lung cancer, particularly in advanced scenarios, its application faces notable challenges, including the development of multidrug resistance (MDR), severe and potentially life-threatening side effects, and the prohibitive costs associated with chemotherapeutic agents [10].

Ivermectin (IVM) is a macrolide anti-parasite, officially approved by the FDA, and is commonly administered orally to treat river blindness, elephantiasis, and scabies. Clinically, it holds significant importance as a well-tolerated and safe broad-spectrum antiparasitic medication. IVM demonstrates antiviral effects as well as the ability to regulate inflammatory diseases by reducing cytokine levels [11]. Moreover, recent studies have highlighted IVM’s potential to inhibit tumor cell growth and overcome drug resistance by modulating various signaling pathways [12,13]. Given the adverse impact of chemotherapeutics on healthy organs and the challenges associated with chemotherapeutic resistance, it becomes imperative to consider drug repurposing as a strategic approach to surmount these obstacles [10]. Despite the promising anti-cancer effects of IVM, it has poor water solubility, which limits its bioavailability and, therefore, its biological activity [14]. In this case, nanoparticles (NPs) have emerged as a promising strategy for enhancing the bioavailability of drugs [15] and increasing drug delivery efficiency by evading clearance pathways in the lungs [16].

Nanotechnology has unlocked new prospects for controlled and targeted drug delivery for cancer therapy. Compared to conventional formulations, NPs offer several advantages: enhanced tissue targeting due to facile surface functionalization, improved tumor distribution through EPR effects, controlled drug release, the ability to deliver multiple drugs with different chemical properties simultaneously, the ability to evade innate biological impediments, and enhanced pharmacokinetic and pharmacodynamic properties, which all in all leads to better therapeutic efficiency and reduced side effects [17]. Recent research has highlighted the advantages of using inhaled nanoparticle delivery as an efficient drug delivery approach to obtain therapeutic benefits at a very low dose of drugs. Nanotechnology offers benefits such as evading clearance pathways and establishing a sustained release pattern. It also enables targeted delivery of drugs to cancer tissues in the lungs, thereby enhancing treatment efficiency [16].

Liposomes and polymeric nanoparticles are widely utilized as nanocarriers due to their favorable properties. Polymeric nanoparticles composed of natural or synthetic polymers offer superior structural integrity of NPs’ storage stability and sustained release of the encapsulated drugs; therefore, these are considered potential candidates for various biomedical applications, including diagnostic or therapeutic delivery [18,19]. Liposomes, on the other hand, exhibit high biocompatibility, resembling biological membranes and seamlessly integrating with the pulmonary surfactant layer in the lungs [20]. To combine the advantages of liposomes and polymeric nanoparticles, lipid–polymer hybrid nanoparticles (LPHNPs) have been developed. The structure of LPHNPs comprises a polymer core encapsulating the drug, a phospholipid shell ensuring biocompatibility, and an outer layer of stabilizer aimed at enhancing in vivo circulation time and providing steric stabilization [21]. These characteristics have made LPHNPs a promising drug delivery platform, especially for pulmonary delivery [20,22]. As reported in the literature, LPHNPs have been used to deliver various therapeutics, such as small molecules, nucleic acids, or a combination of both, as inhalable formulations for pulmonary diseases and achieved the desired outcome [23,24,25,26,27,28,29,30,31].

Given the potential of IVM as an anti-cancer agent and recognizing the advantages of lipid–polymer hybrid nanoparticles (LPHNPs) for pulmonary delivery, the aim of this study is to develop inhalable IVM-loaded LPHNPs and evaluate their characteristics and suitability for pulmonary delivery. For this purpose, we used (1) polycaprolactone, serving as the polymer core, which provides a sustained release profile, structural integrity, and stability; (2) lecithin, forming the lipid shell to improve biocompatibility by mimicking the surfactant lining of the lung; along with (3) Pluronic F127 as the outer stabilizer/surfactant. The ability of Pluronic F127 to attenuate the binding of NPs to mucin and increase mucus penetration has been reported in previous studies [32,33]. Therefore, for pulmonary delivery, where mucus presents a barrier to nanoparticle transport, using such a stabilizer in developing drug delivery vehicles can enhance pulmonary delivery efficiency. Developed formulations were evaluated in terms of physicochemical characteristics, entrapment efficiency and drug loading, release profile, solid phase characteristics, flow, and aerosolization properties to achieve an optimized DPI formulation.

To the best of our knowledge, LPHNPs have never been studied for delivering IVM through the pulmonary route as a DPI formulation. Therefore, developed powder formulations were characterized properly to determine their suitability for pulmonary delivery as a DPI formulation.

## 2. Materials and Methods

### 2.1. Materials

Ivermectin, Pluronic^®^ F127, polycaprolactone (Mn 70,000–90,000 g/mol), and dialysis bags (12,000 Da) were purchased from Sigma-Aldrich, St. Louis, MO, USA. Soybean lecithin was obtained from Merck Millipore, Bangalore, India. HPLC-grade acetonitrile and methanol were supplied from RCI Labscan, Bangkok, Thailand and Fisher Chemical, Couva, Trinidad, respectively. Dichloromethane and ethanol were provided from Thermo Fisher Scientific, Waltham, MA, USA. Deionized double distilled water (Milli-Q water) was used in all experiments.

### 2.2. IVM-Loaded LPHNP Preparation

IVM-loaded LPHNPs were prepared using a single-step emulsion solvent evaporation method with polycaprolactone (PCL) as the polymeric compartment, lecithin as the lipid compartment, and Pluronic F127 as a surfactant (stabilizer). Different formulations of LPHNPs were developed as described by Godara et al. [34] with slight modifications. In this method, appropriate amounts of PCL, lecithin, and IVM were dissolved in dichloromethane (DCM), as indicated in Table 1. This organic phase was added dropwise at a speed of 1 mL/min to an aqueous phase containing different concentrations of Pluronic F127 under continuous stirring at room temperature, followed by homogenization in an ice bath (2–8 °C) at 10,000 rpm for 3 min using a high-speed homogenizer (IKA ULTRA-TURRAX^®^ T25 (Figure 1) (Staufen im Breisgau, Germany). The nanopreparation was then stirred overnight for solvent evaporation. Developed nanoparticle suspensions were centrifuged at 14,000× *g* rpm for 30 min, washed three times with deionized water, and then freeze-dried (Freeze Dryer Alpha 1–4 LD plus (Christ, Osterode am Harz, Germany)) to obtain powder formulation.

### 2.3. Physicochemical Characterization

The particle size, polydispersity index (PDI), and zeta potential of developed LPHNPs before and after freeze drying were determined by dynamic light scattering (DLS) technique using Zetasizer Nano ZS (Malvern Instruments, Worcestershire, UK). For samples before freeze drying, fresh NP suspensions were used for size analysis with appropriate dilution using deionized water [35]. Moreover, freeze-dried powder (3 mg) was suspended in 5 mL of deionized water and sonicated for 15 min before DLS analysis. All measurements were performed in triplicate at room temperature.

### 2.4. Morphology of Nanoparticles

The morphological examination of the developed LPHNPs was observed by scanning electron microscopy (SEM) using Zeiss Sigma Field Emission. Prior to freeze-drying, a 10 μL drop of the LPHNP suspension was applied to a silicon wafer and allowed to air-dry. The freeze-dried LPHNP powder was then separately mounted on an aluminum stub using carbon adhesive tape. Excess particles on the adhesive tape were removed by blowing with nitrogen gas. Both samples were subsequently coated with a conductive sputtered gold layer.

### 2.5. Drug Loading and Entrapment Efficiency

Drug loading and entrapment efficiency were analyzed by HPLC through an indirect method. In this method, the non-entrapped drug (free drug) was collected by centrifugation at 14,000× *g* rpm for 30 min. An aliquot of supernatant was extracted, diluted with methanol (1:20), and filtered by a 0.22 syringe filter to be injected into the system. The amount of IVM in the supernatant was determined with reference to the standard calibration plot. Each sample was repeated 3 times. After achieving the amount of IVM, drug loading and entrapment efficiency were calculated using Equations (1) and (2) below:(1)$$Entrapment\;efficiency\left(\%\right)=\frac{total\;drug-free\;drug}{total\;drug}\times 100$$
(2)$$Drug\;loading\left(\%\right)=\frac{total\;drug-free\;drug}{nanoparticle\;weight}\times 100$$

### 2.6. In Vitro Drug Release Study

To examine the release behavior of developed IVM-loaded NPs, the dialysis bag technique was employed in triplicate, followed by a previously reported method with little modifications [35]. To prepare the dialysis membrane bag (MW 12,000 Da, Sigma-Aldrich, St. Louis, MO, USA) for the release study, it was immersed in deionized water for a duration of 24 h to ensure its readiness and appropriateness for the experimental procedure. A certain amount of IVM-loaded LPHNP formulations (equivalent to 2 mg IVM) was dispersed and placed in a dialysis membrane bag. A dialysis bag was then placed in a beaker containing 50 mL of ethanol/water (50:50) media under continuous magnetic stirring with a speed of 100 rpm at a controlled temperature of 37 °C. At different time intervals (0.5, 1, 1.5, 2, 4, 6, 12, 24, 48, 72, and 96 h), a 1 mL sample was withdrawn from the receiver media and immediately replaced with equal amounts of fresh mixture to keep the media at a constant volume to maintain the sink condition. Samples were analyzed by HPLC at 245 nm.

### 2.7. Kinetics of IVM Release from LPHNPs

Furthermore, the kinetics of IVM release from LPHNPs was mathematically modeled by calculating the correlation coefficient (r) value for each kinetic model, namely zero-order, first-order, Higuchi, and Hixon–Crowell. The Korsmeyer–Peppas model was also used to characterize the release mechanism of IVM from the LPHNP formulation by calculating the release exponent “n”.

### 2.8. Solid-Phase Characterization

#### 2.8.1. Differential Scanning Calorimetry (DSC)

To evaluate the thermal behavior of the formulations, pure IVM, blank NPs, and IVM-loaded NPs were studied by DSC on TA Instruments, model Q100 DSC (New Castle, DE, USA)). Developed powder formulations were accurately weighed (3 ± 0.1 mg) and placed in a hermetic aluminum pan, sealed, and heated. An empty sealed pan was used as a reference. Both pans underwent heating within a specified temperature range of 25–300 °C at a constant heating rate of 10 °C/min.

#### 2.8.2. Thermogravimetric Analysis (TGA)

A NETZSCH Simultaneous Thermal Analyzer (STA) 449 F3 Jupiter was used to evaluate the thermogravimetric behavior and decomposition of pure IVM, blank NPs, and IVM-loaded NPs. For this purpose, 5–8 mg of IVM, blank NPs, and IVM-loaded NPs were placed in TGA alumina crucibles and heated with a heating rate of 10 °C/min from 50 to 800 °C. An empty alumina crucible was used as the reference.

#### 2.8.3. Attenuated Total Reflection–Fourier Transform Infrared (ATR–FTIR)

ATR-FTIR spectroscopy was performed using a Thermo is5 FTIR spectrometer (Nicolet, Madison, WI, USA) to analyze the chemical composition and structure of the nanoparticles. A small amount of powder samples was placed on top of the diamond crystal and secured with a high-pressure clamp. Spectra were obtained within the range of 400−4000 cm^−1^ with a resolution of 8 cm^−1^ and 64 scans. Data were analyzed using OMNIC 8.0 software.

#### 2.8.4. Powder X-ray Diffraction (PXRD)

X-ray diffraction measurements—capillary transmission

IVM, Pluronic F127, and lecithin were measured in capillary (internal diameter 0.8 mm) transmission geometry using a Rigaku SmartLab X-ray diffractometer (Tokyo, Japan). A focusing Goebel mirror in a CBO-E module was used to converge the X-ray beam from a Cu X-ray tube (λ = 1.54059 Å, 40 kV 40 mA), followed by a height limiting slit of 15 mm. Soller slits of 2.5° were used on both primary and secondary beam paths. A Hypix3000 detector (Rigaku, Tokyo, Japan) collecting diffraction signals in 1D mode with a PSD opening of 20 mm after an extended 6.6 mm anti-scattering slit and a 12 mm receiving slit. The capillary samples were spun at 15 rpm during XRD pattern collection from 3 to 70 °2θ at 0.02° step size in 1 h.

X-ray diffraction measurements—foil transmission

The polycaprolactone polymer beads were melted into a piece of self-standing foil (0.8 mm thickness), and their XRD pattern was taken in foil transmission geometry. IVM-loaded LPHNPs were held between two Kapton foils and measured in foil transmission geometry. The blank Kapton foil background was also collected. Both foil transmission XRD data were collected using the same focusing X-ray beam optics and the same measurement scheme described for capillary transmission.

### 2.9. Particle Density and Flow Property

The flow properties of the developed powder were determined using Carr’s index (CI), Hausner ratio (HR), and angle of repose (θ) according to the relevant equations (Equations (3) and (4)). The bulk density and tapped density of the nanoparticle powder were measured using a graduated cylinder in a tapped density tester (ERW-SVM101202, ERWEKA, Langen, Germany). Certain amounts of NP powder (500 ± 0.5 mg) were placed in a 5 mL graduated cylinder to record the initial volume (V_0_). The cylinder was then subjected to 500 mechanical taps in the density tester to establish the new volume (V_1_). Using V_0_ and V_1_, Carr’s index and the Hausner ratio were calculated according to Equations (3) and (4) [35]. Each measurement was performed in triplicate.
(3)CI=100[(plain volume−tapped volume)/plain volume]
(4)Hr=tapped density/bulk density

The angle of repose is a crucial indicator for evaluating the flow characteristics of nanoparticle powders. It defines the maximum angle relative to the horizontal plane of a conical heap of particles. To measure the angle of repose, 250 ± 0.5 mg of nanoparticle powder was gradually poured through a funnel into a beaker situated roughly 3 cm beneath the funnel’s tip. Once the particles settled, the height (h) and base diameter (d) of the resulting cone were recorded. The angle of repose was then calculated using these measurements [35] according to the following Equation:(5)θ=tan−1(2h/d)

### 2.10. In Vitro Aerosolization Study

Aerosolization performances of the developed NP powder formulations were evaluated by a twin-stage impinger (TSI) following the protocol outlined in the British Pharmacopeia. A Breezehaler^®^ (Novartis Pharmaceuticals Pvt Ltd., Macquarie Park, NSW, Australia) was used as the DPI device. Then, 7 mL and 30 mL of water/ethanol 50:50 were poured into stage 1(S1) and stage 2 (S2) of the TSI, respectively. The airflow through the TSI was set to 60 L/min, regulated by a vacuum pump (D-63150, Erweka, Langen, Germany), and monitored through a calibrated digital flow meter (Fisher and Porter, Model 10A3567SAX, London, UK).

In the context of aerosol characterization studies, 20 ± 0.5 mg of the nanoparticle powder samples were loaded into size 3 hypromellose capsules (Vcaps^@^ Plus, Capsugel, Lonza, Basel, Switzerland). These capsules were then placed in a Breezehaler^®^ dry powder inhaler (DPI) and twisted using the DPI device. Actuation of the apparatus was performed by the vacuum pump for 5 s at 60 ± 5 L/min to disperse the powder formulations in different stages of the TSI device. This procedure was conducted 5 times for each formulation (*n* = 5). Following each experimental run, all stages of TSI underwent separate washing with ethanol/water (50:50), and the quantity of IVM was determined by both HPLC assay and gravimetric analysis.

A validated method developed in our laboratory was employed for the gravimetric analysis. For this analysis, filter paper (orifice 0.20 μm, Phenomenex, Torrance, CA, USA) that had been dried and weighed was utilized to filter washings from each stage of the TSI device. After filtration, the particles that had accumulated on the filter paper were dried at 60 °C for 24 h until the filter paper reached a constant weight. This weight was then used to gravimetrically determine the mass of NPs. For chromatographic analysis, washings from each stage were gently stirred (100 rpm) for 96 h at 37 °C to ensure drug was released. The HPLC method was then used to measure the amount of IVM. Thus, both gravimetric and chromatographic methods contributed to determining the quantity of IVM deposited into stage 2 from the IVM-loaded LPHNPs [36].

Aerosolization performance of the prepared formulations was determined by measuring recovered dose (RD), emitted dose (ED), and fine particle fraction (FPF) using Equations (6) and (7). RD is the total amount of particles collected from the inhaler, S1 and S2. ED is the fraction of RD delivered from the inhaler into S1 and S2. FPF is defined as the fraction of RD deposited in the S2 of TSI.
(6)$$ED=\frac{S1+S2}{RD}\times 100$$
(7)$$FPF=\frac{S2}{RD}\times 100$$

### 2.11. HPLC Assay

IVM analytical assay was carried out using HPLC, as previously reported [37]. For this purpose, the calibration plot was produced using a 1 mg/mL Ivermectin stock solution by dissolving 5 mg Ivermectin powder in 5 mL methanol in a volumetric flask. Stock solution was diluted with methanol to achieve various concentrations (2.5, 5, 10, 25, and 50 µg/mL) for developing a calibration plot. For HPLC analysis, an Agilent HPLC Series 1100 (Santa Clara, CA, USA) was used to perform the analytical experiment. A mixture of acetonitrile (53.0%), methanol (27.5%), and ultra-pure water (19.5%) was used as the mobile phase and a Varian Microsorb 100 C18 column (4.6 × 250 mm) as the stationary phase (Palo Alto, CA, USA). Solvents were mixed properly, filtered through 0.22 µm filters, and degassed by putting them in ultrasonic bath for 10 min. Flow rate and injection volume were set at 2 mL/min and 20 μL, respectively, with a detection wavelength set at 245 nm. A linear plot of the area under the curve (AUC) versus concentration with a coefficient of determination was obtained with limit of detection (LOD) and limit of quantification (LOQ) of 0.60 and 1.83 µg/ml, respectively, calculated based on the calibration curve [38].

### 2.12. Statistical Analysis

Each experiment was conducted in triplicate, and the results are presented as mean ± SD. Statistical differences between samples were assessed using one-way ANOVA followed by Tukey’s post hoc test, with significance set at *p* < 0.05. This analysis was performed using GraphPad Prism software, version 10.0.2.

## 3. Results and Discussion

### 3.1. IVM-Loaded LPHNP Preparation

To evaluate the effect of lipid/polymer ratio and effect of concentration of stabilizer on various characteristics of NPs, including size and entrapment efficiency, three different lipid amounts and three different surfactant concentrations were considered for this study, as described in Table 1. Formulated nanoparticle suspensions were centrifuged at 14,000× *g* rpm for 30 min and washed three times with water. Finally, NPs in the form of powder were obtained by freeze drying using Freeze Dryer Alpha 1–4 LD plus.

### 3.2. Physicochemical Characterization

#### 3.2.1. Particle Size and Size Distribution

The optimization of formulation development necessitates the precise modulation of physicochemical parameters, particularly the mean values of size and size distribution. Mean particle size and size distribution of developed IVM-loaded LPHNPs were obtained from the dynamic light scattering (DLS) before and after freeze drying to evaluate the effect of freeze drying on formulations. As presented in Table 2, before freeze drying, the average particle size of NPs was in the range of 303–350 nm with a polydispersity index of less than 0.5, which demonstrated uniformity and narrow size distribution (Figure 2). A significant reduction in particle size was observed from F1 to F2 (*p* < 0.01) and F3 (*p* < 0.0001) with increasing the ratio of lecithin. A similar result was achieved in a previous study by Ren et al. [39]. This can be explained by the stabilizing effect of lecithin and its ability to provide steric hindrance [30]. Moreover, particle size decreases significantly (*p* < 0.05) when the concentration of Pluronic^®^ F127 increases, but beyond a certain point (0.5%), an additional increase in concentration leads to a smaller particle size, which is in accordance with a previous study on LPHNPs [31]. It can be concluded that at a lower concentration (0.25%), there may have been an inadequate quantity of surfactant to cover the LPHNPs, and increasing the concentration of surfactant to 0.5% allows for proper orientation of surfactant molecules, which leads to a reduction in interfacial tension between the solvent phases and, therefore, smaller particle size [40]. However, at a higher concentration of surfactant, excess surfactant molecules fuse with the nanoparticles, and this adsorption results in an increasing average particle size of LPHNPs [41,42]. After freeze-drying, particle size increased due to the formation of agglomerates, which is a common phenomenon during the freeze-drying process [43], but the size trend was similar to before freeze-drying.

#### 3.2.2. Zeta Potential

The physical stability of nanoparticles is significantly influenced by the zeta potential of the prepared nanoparticles. This is an indicator of the degree of repulsion between particles that bear similar charges, offering a predictive measure to ensure the enduring stability of the nanoparticles [34]. As reported in the literature [44,45], zeta potentials of higher than −30.0 mV or + 30.0 mV are required for good physical stability as they provide enough repulsive forces between NPs and prevent aggregation. The zeta potential values of developed formulations were higher than −30 mV before and after freeze-drying (Table 2), indicating the proper stability of the developed formulations. This negative zeta potential could be due to the anionic characteristics of soybean lecithin, causing repulsion between nanoparticles and limiting aggregation [40].

### 3.3. Particle Shape and Morphology

Examination under SEM revealed images of the formulated IVM-loaded LPHNPs displaying well-defined spherical particles with smooth surfaces before freeze-drying (Figure 3A,B). After freeze-drying, agglomerated NPs were observed along with some individual particles (Figure 3C,D). The agglomeration of NPs is a common phenomenon after freeze-drying. Nevertheless, agglomerated particles were all in an acceptable range for inhalation (less than 5 µm) [43].

### 3.4. Entrapment Efficiency and Drug Loading

Entrapment efficiency and drug loading of IVM-loaded LPHNPs were calculated and presented in Table 3. F1 showed the highest EE of 80.59%, which is significantly more than all other formulations (*p* < 0.05). Table 3 shows that EE is directly related to particle size. As reported previously, smaller particles have lower EE as they have a lower capacity to entrap the drug inside the particle [21,46]. However, although F1 and F5 have similar particle sizes, EE in F5 is significantly lower (*p* < 0.0001). This could be due to the partitioning of the drug in the presence of higher concentrations of surfactant. High concentrations of surfactant lead to the partitioning of the drug from the organic to aqueous phase and, therefore, drug solubilization [47,48]. Developed formulations also revealed drug loading of 7–10%, which is in accordance with previous studies working on nanoparticulate systems for drug delivery with DL of less than 10% [49,50].

### 3.5. In Vitro Drug Release Study and Kinetic of Release

Controlled and sustained drug delivery is a valuable feature for nanoparticle-based drug delivery systems as it leads to reduced dose frequency and, therefore, enhanced patient compliance. Figure 4 illustrates the cumulative percentage of IVM release from IVM-loaded LPHNPs and IVM suspension. From Figure 4, it can be observed that all formulations revealed a slow-release pattern within 96 h. None of the developed LPHNPs showed any initial burst drug release before one hour from the start of the experiment, which indicates that IVM is successfully encapsulated inside LPHNPs rather than adhering to the surface of LPHNPs [51]. F1 showed the latest onset of IVM release starting after two hours of the test. However, F2, F3, and F4 released IVM faster after 1.5 h of the release experiment. The faster release of IVM from these formulations could be explained by the smaller sizes of NPs, as smaller particles provide a higher surface area for the drug release [52]. On the other hand, although F5 had a similar particle size to F1, it showed the fastest onset of IVM release among all formulations starting from the first hour of the release evaluation, which could be related to the effect of Pluronic^®^ F127, as F5 had the highest concentration of Pluronic F127, which leads to a reduction in interfacial tension [53] (Table 1). Previously, Tahir et al. studied methotrexate-loaded LPHNPs with different concentrations of Lutrol^®^ F-68 as surfactant and observed similar findings [47]. The release test was continued until all formulations reached a steady state of IVM concentration after 96 h. At the end of the experiment, F1, F3, and F5 achieved similar results (*p* > 0.05), with a maximum of approximately 60% cumulative release IVM, which is in agreement with the IVM release from IVM-loaded SLNs [35]. However, F2 and F4 showed lower IVM release over this time compared to other formulations, with around 52% and 55%, respectively. Although the difference was not significant (*p* > 0.05), the reason behind this phenomenon could be that, compared to F1, all other formulations had more excipients and, therefore, a thicker layer was covering the polymeric core where the drug is encapsulated [54]. For example, F2 and F3 had higher amounts of lipid, while F4 and F5 had higher concentrations of Pluronic F127, covering the polymer core as a thicker layer and limiting the drug release over time, but since F3 had the smallest size among all formulations, the surface area provided by smaller particles probably allowed an increased interaction with the solvent and an easier release compared to F2 or F4. Moreover, the surfactant activity of Pluronic F127 in F5, with the highest concentration, helps solubilize IVM and thus increase release over 96 h.

### 3.6. Kinetics of IVM Release from LPHNPs

A mathematical kinetic model was utilized to analyze the release mechanism of IVM from the developed LPHNPs. To elucidate the mechanism of IVM release from the LPHNPs, the in vitro drug release data for IVM-loaded LPHNPs were fitted into various kinetic models of drug release (zero-order, first-order, Higuchi, Hixson–Crowell, and Korsmeyer–Peppas) and the model that best fit the data was identified by selecting the one with the highest *r*^2^ values (Table 4). The zero-order model illustrates systems where the drug release rate is independent of drug concentration. In contrast, the first-order model describes systems where the drug release rate is dependent on the drug concentration. Higuchi’s model refers to the liberation of the drug from an insoluble matrix as a process dependent on the square root of time, which is based on Fickian diffusion. The Hixson–Crowell model characterizes a system in which the cube root of the quantity of drug released linearly correlates with time, applicable particularly to systems where the surface changes over time. The Korsmeyer–Peppas model is another kinetic model that can generally characterize various release phenomena, including those driven by either diffusion or erosion processes [55,56]. Release data presented in Table 4 showed that all formulations were properly fitted with the Higuchi model (*r*^2^ = 0.86–0.91). Thus, it can be interpreted that diffusion is a possible mechanism of IVM release from developed LPHNPs. For further comprehension of the release mechanism, the Korsmeyer–Peppas model was also utilized to determine the transport model based on the release exponent (n). As reported in the literature, if *n* ≤ 0.45, it suggests Fickian diffusion, while if 0.45 < *n* < 0.89, it indicates anomalous transport, and if *n* > 0.89, it is indicative of super case II transport [57]. The *n* value for F1 was 0.9112, indicating the drug release might follow super case II transport, and thus erosion is the main mechanism for IVM release in this formulation. However, since F2–F5 showed *n* values between 0.45 and 0.89, these formulations follow anomalous transport and, therefore, diffusion and erosion are both contributing to the IVM release in these formulations. Similar behavior was reported by Soomherun et al. while developing nicardipine-loaded LPHNPs [56].

### 3.7. Solid-Phase Characterization

#### 3.7.1. Differential Scanning Calorimetry (DSC)

DSC was utilized to conduct phase transition studies, which determined the crystalline and amorphous states of the developed formulations [47]. As demonstrated in Figure 5, the IVM thermogram showed a sharp endothermic peak at 160.88 °C, corresponding to its melting point and confirming its crystalline structure, which is in accordance with the literature [58]. All five formulations of IVM-loaded LPHNPs showed similar characteristic endothermic peaks to the blank LPHNPs. The absence of IVM endothermic peak in IVM-loaded LPHNPs suggests the successful encapsulation of IVM inside LPHNPs [59]. This result reflects that the encapsulated drug is probably in an amorphous state, which is in line with previous findings that illustrated the conversion of crystallin to amorphous form of the drug while encapsulation in NPs [47]. Furthermore, the DSC thermogram of the developed LPHNPs showed that the polymer’s melting peak, observed at 56.54 °C, did not shift significantly. This indicates an absence of interaction between the drug and polymer, confirming the compatibility of components and drug [47].

#### 3.7.2. Thermogravimetric Analysis (TGA)

The thermal behaviors of IVM, blank LPHNPs, and IVM-loaded LPHNPs were investigated by TGA. The TG curve of IVM displayed in Figure 6 identifies four separate phases of mass loss in IVM. Initially, at around 152.1 °C, mass loss occurs due to water desorption and changes in the crystallinity of the drug [60]. The next phase of thermal decomposition happens at 295.9 °C, attributed to the degradation of C–O–C aliphatic esters, influenced by their spatial configurations. In the third phase, molecular degradation starts at 326.7 °C, leading to the axial deformation of O–H bonds and methyl groups, followed by the disruption of unsaturated lactones. Further decomposition of the sample happens when the temperature rises above 450 °C [60,61]. On the other hand, as can be seen from Figure 6, all developed formulations showed a sharp single-step mass loss at higher temperatures (360–500 °C) compared to free IVM. LPHNPs showed better thermal stability compared to free IVM, as prepared formulations revealed 11–16% mass loss below 360 °C, while more than 63.4% mass loss was observed for free IVM below 360 °C.

#### 3.7.3. Attenuated Total Reflection–Fourier Transform Infrared (ATR–FTIR)

FTIR spectroscopy can be employed to verify the compatibility between drugs and excipients and the entrapment of drug in LPHNPs. Figure 7 demonstrates the infrared spectra for IVM, blank LPHNPs, and IVM-loaded LPHNPs. The FTIR spectrum of IVM displayed characteristic peaks: a saturated aliphatic ketone C=O stretch was identified at 1729.51 cm^−1^. The peak at 1675.28 cm^−1^ corresponded to a double bond in unsaturated lactones. Additionally, ketones presented peaks between 1381.78 and 1311.86 cm^−1^ owing to the C=C group, and aliphatic ethers were evident in the C-O-C stretching range of 1181.32 to 1022.58 cm^−1^. Moreover, the O-H stretching was observed at 3476.17 cm^−1^, along with peaks at 2936.97 cm^−1^, which were indicative of methyl groups associated with C-H stretching [62]. The absence of IVM characteristic peaks in developed formulations showed successful encapsulation of IVM inside developed LPHNPs. Similar results were also observed when developing IVM-loaded chitosan–alginate NPs and IVM-loaded SLNs [35,63]. Moreover, since there has not been a new peak formed in IVM-loaded LPHNPs and the existing peaks have not changed substantially, it can be concluded that there are no substantial interactions between the carrier components and the drug [64].

#### 3.7.4. Powder X-ray Diffraction (PXRD)

The API phase was identified to be Ivermectin hemihydrate ethanolate (ICDD PDF# 00-071-1568). Rietveld structure refinement using a Rigid Body model of the Ivermectin molecule was performed in DIFFRAC.TOPAS v7 (Figure 8A). The monoclinic unit cell (a = 40.826(6) Å, b = 9.264(1) Å, c = 14.891(2) Å, β = 73,136(7) Å) of space group C121 contains four Ivermectin molecules, four ethanol molecules, and two water molecules. The XRD patterns of F1 as the base formulation and excipients are shown in Figure 8B. The main feature of F1 demonstrated the same diffraction signals as PCL, with a minor content of Pluronic F127 peak around 19 °2θ. The low-angle hump near 5 °2θ is from the Kapton foil holding the F1. The lecithin sample is in a total amorphous form and, therefore, does not contribute any crystalline peaks. The disappearance of characteristic peaks of IVM in prepared formulation implies that the encapsulated drug either formed a molecular dispersion or is present as an amorphous state within LPHNPs, confirming the results achieved by DSC and FTIR [64,65].

### 3.8. Particle Density and Flow Property

Powder flow characteristics have a critical role in DPI formulation performance. Carr Index (CI), Hausner Ratio (HR), and angle of repose are three parameters used to evaluate the flowability of powder formulations. Following bulk density and tapped density measurement, values for CL and HR were measured and are presented in Table 5. A CI value under 25% indicates favorable flow properties, while a CI exceeding 25% denotes poor flowability, which is typical in cohesive powders [43]. On the other hand, an HR less than 1.25 indicates the proper flowability of powder, whereas an HR greater than 1.25 points to suboptimal flowability [36]. All developed formulations showed promising flow properties with CI values between 12.25 and 23.96, as well as HR ranging from 1.14 to 1.32 (Table 5). Apart from CI and HR, the angle of repose is also an indicator of powder flow characteristics, where angle of repose values higher than 40° represent more powder cohesion and, therefore, poor flow properties [66]. Table 5 also demonstrates the angle of repose data for developed LPHNPs. θ values were less than 40 for all formulations, which is in accordance with the other two flowability indicators. All formulations showed promising flow properties, with F1 showing the lowest (*p* < 0.05) compared to all formulations except blank and F3, demonstrating the potential of LPHNPs to provide the desired dispersibility behavior as DPI formulation.

### 3.9. In Vitro Aerosolization Study

The aerosolization properties of blank and IVM-loaded LPHNPs were assessed based on RD, ED, FPF, and FPD. As represented in Table 6, all formulations showed proper dispersion of particles into various stages of TSI, with RD values ranging from 95.19 to 98.44%. Moreover, the EDs for all developed formulations were in the range of 87.59 to 90.55%, which indicates the efficient emission of formulations from the device. Successful emission could be explained as a result of the spherical shape of developed NPs according to the SEM images. As reported in the literature, particles with spherical shapes show less adhesion to the surface of the inhaler; thus, appropriate ED% can be achieved [67]. As discussed earlier (Table 2), all formulations had high negative zeta potential, which prevents the aggregation of NPs by providing proper repulsive forces among particles and thus could result in FPFs ranging from 18.53 to 24.77%. Among all formulations, F1 showed the highest FPF of 24.77% (*p* < 0.05) compared to all formulations (except blank and F3), which could be predicted from the flow properties of this formulation showing the lowest CI, HR, and θ. The amount of IVM delivered to stage two of TSI was 0.31–0.43 mg (from 1.5–2 mg of loaded IVM) with developed NPs, so it is anticipated that this amount of IVM would reach deep regions of the lung and be absorbed. Further in vitro cell culture studies will also be conducted to evaluate the effect of IVM-loaded LPHNPs on lung cancer cell lines and determine the cellular uptake and cell-killing ability of developed formulations. However, animal studies are warranted to examine the safety and in vivo efficacy of the developed formulations in the lungs. Various methods have been used for the pulmonary delivery of drugs in animal models. Intratracheal administration is one of the methods used for pulmonary delivery, where the drug is delivered through a thin tube inserted in the intratracheal region. Whole-body exposure is another method where the drug is absorbed through the entire body surface, leading to potentially complicated dosimetry. Nose-only inhalation devices, on the other hand, allow the drug to be uniformly deposited into the lungs through inhalation ports using a DPI device [68,69,70]; however, animal studies are beyond the scope of this manuscript.

## 4. Conclusions

This research presents an effective and viable approach for developing inhalable formulations of IVM-loaded LPHNPs for pulmonary delivery against critical pulmonary diseases, including lung cancer. Developed formulations demonstrated desirable physicochemical properties such as nano-ranged particle size, narrow size distribution, and high zeta potential. ATR-FTIR and DSC studies confirmed the successful encapsulation of IVM inside LPHNPs with an entrapment efficiency of approximately 68–80%, resulting in a sustained release profile over 96 h, potentially leading to an extended duration of action. Prepared formulations illustrated good flowability and aerosolization properties with FPF values between 18.5 and 24.87%, indicating LPHNPs as promising candidates for dry powder inhalation, which could overcome the limitations of the currently available oral dosage form. However, further research is necessary to examine the therapeutic efficiency and safety of LPHNPs for pulmonary delivery.

## Figures and Tables

**Figure 1 pharmaceutics-16-01061-f001:**
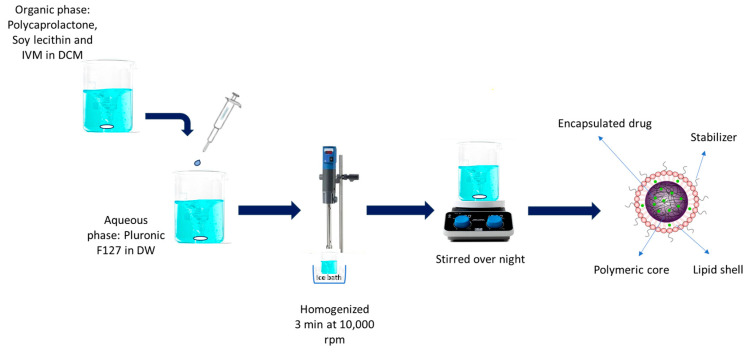
A schematic diagram of preparation process of LPHNPs.

**Figure 2 pharmaceutics-16-01061-f002:**
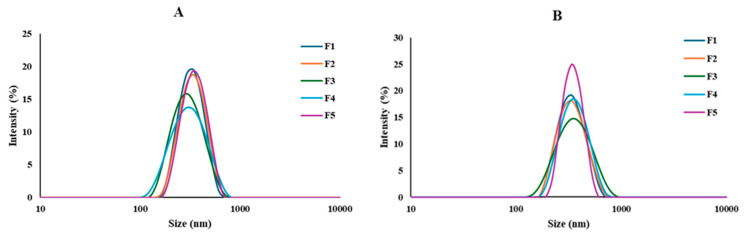
Particle size distribution of the developed formulations (**A**) before freeze-drying and (**B**) after freeze-drying (mean ± SD; *n* = 3).

**Figure 3 pharmaceutics-16-01061-f003:**
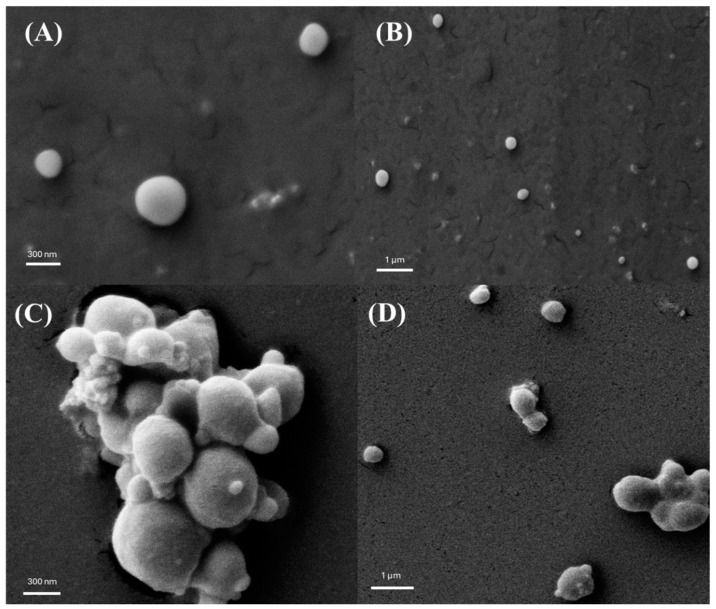
SEM images of the developed NPs (**A**,**B**) before and (**C**,**D**) after freeze-drying.

**Figure 4 pharmaceutics-16-01061-f004:**
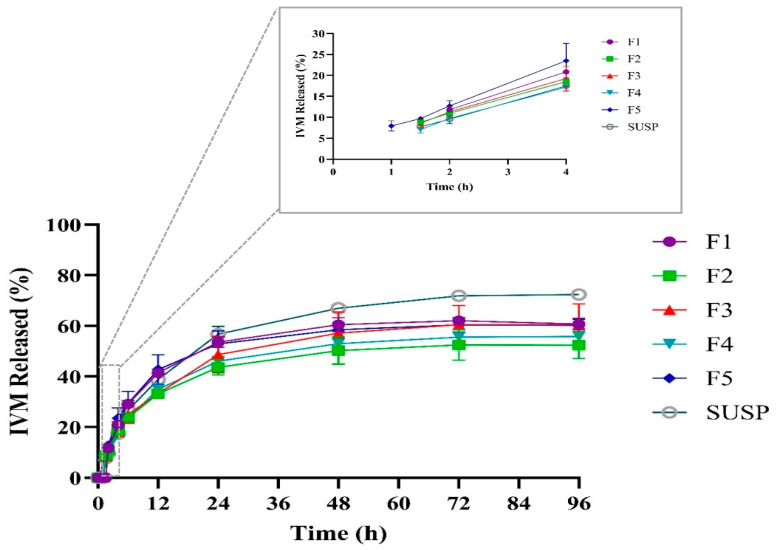
Release profile of IVM from the prepared LPHNPs over 96 h and initial release in the first 4 h presented in top right corner.

**Figure 5 pharmaceutics-16-01061-f005:**
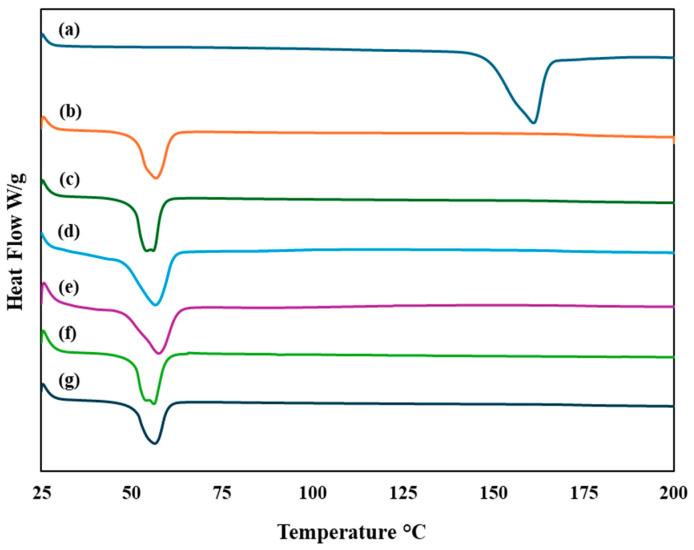
DSC thermograms of (a) pure IVM, (b) blank LPHNPs, and (c–g) IVM-loaded LPHNPs F1–F5 (exothermic up).

**Figure 6 pharmaceutics-16-01061-f006:**
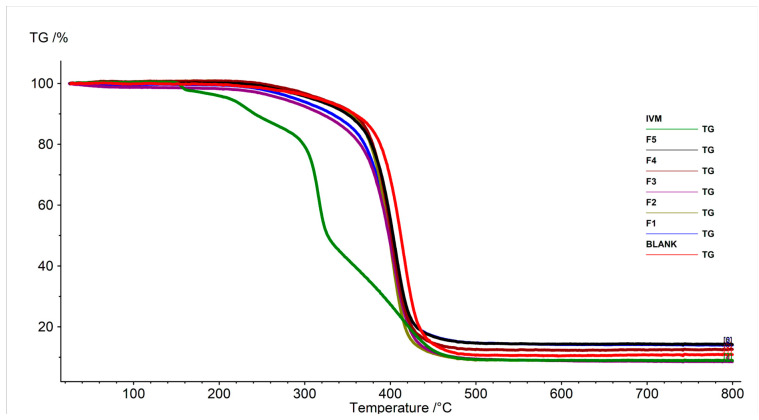
TGA thermograms of pure IVM, blank LPHNPs, and IVM-loaded LPHNPs (F1–F5).

**Figure 7 pharmaceutics-16-01061-f007:**
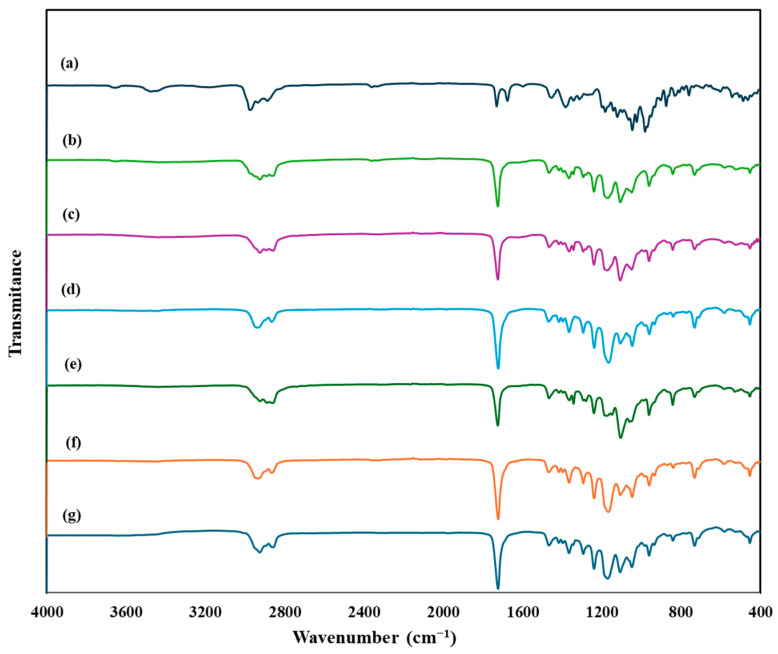
FTIR spectra of (a) pure IVM, (b) blank LPHNPs, and (c–g) IVM-loaded LPHNPs F1–F5.

**Figure 8 pharmaceutics-16-01061-f008:**
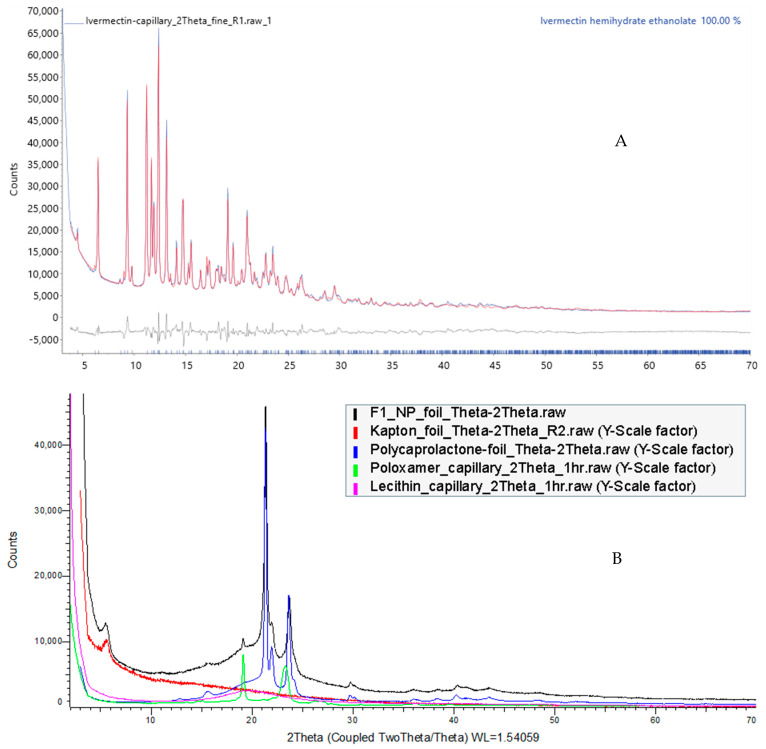
XRD patterns of (**A**) pure IVM, (**B**) PCL, lecithin, Pluronic F127 (poloxamer), and IVM-loaded LPHNPs.

**Table 1 pharmaceutics-16-01061-t001:** Formulation composition of developed NPs.

Formulation Code	PCL (mg)	Lecithin (mg)	Pluronic F127 (%*w*/*v*)	IVM (mg)
F1	45	30	0.25	6
F2	45	45	0.25	6
F3	45	60	0.25	6
F4	45	30	0.5	6
F5	45	30	0.75	6

**Table 2 pharmaceutics-16-01061-t002:** Particle size, PDI, and zeta potential of developed formulations (data presented as mean ± S.D., *n* = 3).

	Before Freeze-Drying			After Freeze-Drying		
	Particle Size (nm)	PDI	Zeta Potential (mV)	Particle Size (nm)	PDI	Zeta Potential (mV)
F1	350.70 ± 16.04	0.32 ± 0.06	−32.03 ± 1.36	438.39 ± 11.26	0.38 ± 0.01	−36.92 ± 0.76
F2	320.34 ± 14.81	0.28 ± 0.04	−34.16 ± 1.25	382.12 ± 13.47	0.34 ± 0.03	−38.38 ± 1.03
F3	302.19 ± 19.27	0.32 ± 0.09	−33.47 ± 1.63	346.83 ± 14.75	0.33 ± 0.04	−37.24 ± 0.82
F4	323.01 ± 7.24	0.42 ± 0.07	−37.98 ± 1.94	368.60 ± 5.39	0.39 ± 0.02	−43.15 ± 0.88
F5	350.75 ± 27.57	0.38 ± 0.09	−38.82 ± 1.69	465.17 ± 28.09	0.44 ± 0.04	−43.8 ± 0.70

**Table 3 pharmaceutics-16-01061-t003:** Entrapment efficiency and drug loading of IVM-loaded LPHNPs (mean ± SD; *n* = 3).

Formulation Code	Entrapment Efficiency %	Drug Loading %
F1	80.59 ± 0.93	10.15 ± 0.22
F2	75.40 ± 1.56	9.23 ± 0.24
F3	71.65 ± 1.54	8.23 ± 0.25
F4	74.64 ± 2.01	8.02 ± 0.35
F5	68.32 ± 1.62	7.38 ± 0.31

**Table 4 pharmaceutics-16-01061-t004:** Kinetic analysis of IVM-loaded LPHNP release data.

Formulation	Zero-Order	First-Order	Higuchi	Hixson–Crowell	Korsmeyer–Peppas	
	*r* ^2^	*r* ^2^	*r* ^2^	*r* ^2^	*r* ^2^	*n*
F1	0.6790	0.7517	0.8632	0.7284	0.8088	0.9112
F2	0.7143	0.7778	0.8929	0.7573	0.7935	0.7858
F3	0.7552	0.8285	0.9191	0.8054	0.8108	0.8108
F4	0.7296	0.7950	0.9025	0.7739	0.8244	0.8138
F5	0.6806	0.7541	0.8723	0.7304	0.7763	0.8146

**Table 5 pharmaceutics-16-01061-t005:** Flow properties of blank and IVM-loaded LPHNPs (mean ± SD; *n* = 3).

	Blank LPHNP	F1	F2	F3	F4	F5
Bulk density	0.21 ± 0.00	0.23 ± 0.01	0.19 ± 0.00	0.27 ± 0.01	0.15 ± 0.01	0.16 ± 0.01
Tapped density	0.25 ± 0.00	0.27 ± 0.01	0.23 ± 0.01	0.33 ± 0.01	0.20 ± 0.00	0.20 ± 0.00
CI	14.37 ± 2.3	13.31 ± 1.69	18.94 ± 3.43	16.37 ± 0.51	22.36 ± 3.88	20.86 ± 2.21
HR	1.17 ± 0.03	1.16 ± 0.01	1.24 ± 0.06	1.19 ± 0.01	1.29 ± 0.07	1.26 ± 0.03
θ	29.52 ± 1.89	28.26 ± 1.23	35.37 ± 0.95	31.31 ± 1.81	37.52 ± 0.91	36.99 ± 0.91

**Table 6 pharmaceutics-16-01061-t006:** In vitro evaluation of particle deposition from blank and IVM-loaded LPHNPs (mean ± SD; *n* = 5).

Formulation	RD%	ED%	FPF%	FPD (µg)
Blank LPHNP	95.20 ± 1.79	90.13 ± 0.59	24.26 ± 0.70	-
F1	98.44 ± 1.16	90.55 ± 0.51	24.77 ± 1.69	438.09 ± 61.34
F2	95.19 ± 1.82	87.59 ± 0.77	19.81 ± 1.19	331.01 ± 26.62
F3	98.18 ± 0.97	89.90 ± 1.32	22.45 ± 1.83	410.68 ± 48.65
F4	97.53 ± 0.84	87.79 ± 0.83	18.53 ± 0.87	314.32 ± 35.76
F5	96.76 ± 0.59	89.66 ± 0.40	21.70 ± 1.51	379.86 ± 33.19

RD, recovered dose; ED, emitted dose; FPF, fine particle fraction; FPD, fine particle dose.

## Data Availability

All relevant data are available in this article.

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
