# Peer review of "Inhaled Ivermectin-Loaded Lipid Polymer Hybrid Nanoparticles: Development and Characterization"

_pharmaceutics, 2024, doi:10.3390/pharmaceutics16081061_

Round 1

Reviewer 1 Report

Comments and Suggestions for Authors

The paper submitted by Kassaee et el. investigates the preparation and characterization of ivermectin-loaded lipid polymer hybrid nanoparticles. The manuscript is clear, well written and the conclusions are supported by the results. However, some corrections are needed:

1. the introduction section can be completed with a recent review reference: https://doi.org/10.3390/polym16020206

2. add units for the Mn of polycaprolactone

3. in order to avoid the agglomeration of the particles during freeze-drying the use of a cryo-protector is mandatory. 

4. it would have been expected that the drug would not be visible in DSC thermograms as the quantity is quite low.

5. as shown in fig 7 and contrary to the statement of the authors, it seems that there are some shifts of the characteristic peaks of LPHNPs after the loading of IVM. F2, F4 and F5 have a different peak around 1200 cm-1 as compared to blank sample. The authors must revise their discussion as the slightest shift is indicative of some interactions between the polymer matrix and drug.

6. cytotoxicity tests must be added. 

Author Response

Q1: the introduction section can be completed with a recent review reference: https://doi.org/10.3390/polym16020206

Response: agreed and the introduction section has been revised accordingly. Please refer to the 5th paragraph of the introduction.

Q2: add units for the Mn of polycaprolactone:

Response: agreed, the unit (g/mol) was added to the revised manuscript.

Q3: in order to avoid the agglomeration of the particles during freeze-drying the use of a cryo-protector is mandatory.

Response: we agree that cryoprotectants are beneficial for making stable NPs. However, many nano-based formulations have been developed without any cryoprotectant in recently published papers:

  • https://doi.org/10.1016/j.jddst.2023.105175
  • https://doi.org/10.1016/j.foodhyd.2023.109264
  • https://doi.org/10.1208/s12249-023-02551-6

Moreover, in our study, agglomerated particles are nanosized and acceptable for pulmonary drug delivery, therefore cryoprotectant was not used in this study. Previous studies in our group also did not use any cryoprotectant for freeze drying but achieved favourable properties for developed formulations:

  • https://doi.org/10.1016/j.ejpb.2016.12.023
  • https://doi.org/10.1016/j.apt.2018.08.011
  • https://doi.org/10.1016/j.ejpb.2020.07.011
  • https://doi.org/10.1371/journal.pone.0261720
  • https://doi.org/10.3390/ph15101223
  • https://doi.org/10.3390/ijms24054532

Q4: it would have been expected that the drug would not be visible in DSC thermograms as the quantity is quite low.

Response: Actually, this is unclear as various studies developed nanoparticles containing high or low amounts of drugs showed no visible peaks for DSC.

For example, using low amounts of drugs, previously published papers demonstrated similar DSC results as we find. Refer to the published papers linked below:

  • https://doi.org/10.1016/j.jddst.2023.105175,
  • https://doi.org/10.1007/s12247-023-09794-7,
  • https://doi.org/10.1016/j.ijbiomac.2018.10.005
  • https://doi.org/10.1016/j.colsurfb.2018.03.052
  • https://doi.org/10.3390/pharmaceutics14010185,
  • https://doi.org/10.3390/ph14080786,

Similarly, the NPs containing high amounts of drug did not show visible DSC peaks. Please refer to the published papers presented below:

  • https://doi.org/10.1021/acsbiomaterials.1c00066
  • https://doi.org/10.3390/pharmaceutics14040884
  • https://doi.org/10.1016/j.colsurfb.2012.04.027
  • https://doi.org/10.1016/j.jddst.2022.103527

Q5: as shown in fig 7 and contrary to the statement of the authors, it seems that there are some shifts of the characteristic peaks of LPHNPs after the loading of IVM. F2, F4 and F5 have a different peak around 1200 cm-1 as compared to blank sample. The authors must revise their discussion as the slightest shift is indicative of some interactions between the polymer matrix and drug.

Response: We superimposed the FTIR figures (refer to the highlighted peaks at 1200cm-1 of the attached file) of all formulations and found no significant changes/ shifts of the peaks in blank and drug-loaded formulations.   Therefore, it can be concluded that there are no substantial interactions. Please refer to the last sentence of section 3.7.3.  

Q6: cytotoxicity tests must be added.

Response: Thank you for this comment. The cytotoxicity and cell culture experiments are ongoing and will be presented in our future paper. This manuscript mainly focuses on the development and physicochemical characterization of the formulations. However, the term “against lung cancer” has been removed from the text and lung cancer has been mentioned as an example of chronic pulmonary diseases and a possible indication for the developed formulation, as ivermectin has proven to be effective against cancer demonstrated in the published articles (below).

  1. https://doi.org/10.1016/j.phrs.2020.105207
  2. https://doi.org/10.3389/fphar.2021.717529
  3. https://doi.org/10.1007/s00280-020-04041-z

Reviewer 2 Report

Comments and Suggestions for Authors

The project focuses on synthesizing nanoparticle powder containing ivermectin for the purpose of treating lung cancer. The essay appears to be well-written and coherent. Nevertheless, a correction is necessary.

1) Figure 1. An ice bath is necessary for the homogenization procedure. Kindly provide a schematic illustrating the setup of an ice bath. It is necessary to provide the temperature for each stage in order to demonstrate the regulation of temperature throughout the production process.
2) In Figure 1, the authors said that ivermectin is enclosed inside the polymer core, but not within the lecithin layer. Is there any rationale for this, considering that ivermectin is a medicine that does not dissolve in water? Shouldn't it also be distributed in this layer?
3) The technique states the use of statistical analysis, however no findings are shown to support this claim. It is intended for the analysis of Table 2, Table 3, Figure 4, Table 5, and Table 6. Please perform analysis for these results
4) Table 6. The abbreviation should be specified as a footnote to the table. Does the FPD has the capacity to demonstrate anticancer activity? Please provide a discussion for this area.
5) In order for the powder to reach the deep lung, it must have a powder particle size between 0.5 µm and 5 µm. Could you please provide the measurements of your powder and confirm if they fall inside this specified range?
6) What is the method of powder delivery to the animal? This is quite challenging but good to have this in the discussion.
7) Authors should also discuss your lung deposition study experimental design to Anderson cascade impactor which has more stages to define the location of the lungs.

Author Response

Q1: Figure 1. An ice bath is necessary for the homogenization procedure. Kindly provide a schematic illustrating the setup of an ice bath. It is necessary to provide the temperature for each stage in order to demonstrate the regulation of temperature throughout the production process.

Response: Ice bath was added to the figure 1 and temperatures are added to the revised manuscript

Q2: In Figure 1, the authors said that ivermectin is enclosed inside the polymer core, but not within the lecithin layer. Is there any rationale for this, considering that ivermectin is a medicine that does not dissolve in water? Shouldn't it also be distributed in this layer?

Response: Agreed. Figure 1 has been revised and the drug molecules were added to both layers.

Q3: The technique states the use of statistical analysis, however no findings are shown to support this claim. It is intended for the analysis of Table 2, Table 3, Figure 4, Table 5, and Table 6. Please perform analysis for these results

Response: Statistical analysis was added to the revised manuscript (where appropriate).  P values have been added to any experiments that involved comparison between formulations.

Q4: Table 6. The abbreviation should be specified as a footnote to the table. Does the FPD has the capacity to demonstrate anticancer activity? Please provide a discussion for this area.

Response: Abbreviations were added in the footnote.

Regarding the FPD and anticancer activity, ivermectin is considered a repurposed drug with the potential to be used as an anticancer. However, it still has not been evaluated in humans and therefore the required dose for having an anticancer effect in humans is not determined. The results of our research mostly focus on the development and characterization of a nano-based inhalable formulation and can serve as a basis for future research when the required dose of ivermectin for anticancer effect in humans is determined.

In terms of the antiparasitic effect of ivermectin, the Cmax after oral administration of 6 mg tablets is 38.2ng/ml (https://doi.org/10.1208%2Fs12248-007-9000-9). However, with the developed inhalable formulation, out of 2mg of drug loaded in the developed NPs, a maximum of 4.38 µg (calculated based on in-vitro aerosolization test) would reach into the deep lungs, and this is much higher than that of the concentration obtained after oral administration of 6mg drug. It should be highlighted that the Cmax mentioned above is for antiparasitic effect and there is no clinical data on the anticancer effect of ivermectin. So, preclinical/clinical studies are warranted to understand the inhaled dose of ivermectin and its anticancer activity.    

Q5: In order for the powder to reach the deep lung, it must have a powder particle size between 0.5 µm and 5 µm. Could you please provide the measurements of your powder and confirm if they fall inside this specified range?

Response: Small nanoparticles (100 nm–1 µm) can easily be accumulated in the deep lungs after pulmonary delivery (https://doi.org/10.1016/j.addr.2021.113953) .

Freeze dried powders of all formulations underwent size measurement by DLS (Table 2) and also based on the SEM images (Figure 3C,D), were in the acceptable range for pulmonary delivery. Previously published papers from our lab followed the same methods for measuring the size and achieved favorable inhalable properties (please refer to the following papers):

  • https://doi.org/10.1016/j.ejpb.2016.12.023,
  • https://doi.org/10.1016/j.apt.2018.08.011,
  • https://doi.org/10.1016/j.ejpb.2020.07.011,
  • https://doi.org/10.1371/journal.pone.0261720,
  • https://doi.org/10.3390/ph15101223,
  • https://doi.org/10.3390/ijms24054532 .

Q6: What is the method of powder delivery to the animal? This is quite challenging but good to have this in the discussion.

Response: Powdered drug delivery to the animal lungs is very challenging. In past, nicotine-loaded chitosan nanoparticles were successfully delivered into mouse lungs through a device developed in our lab (ref 70).

 A new paragraph on methods used for pulmonary delivery to animals has been added to the end of the result & discussion section.

Q7: Authors should also discuss your lung deposition study experimental design to Anderson cascade impactor which has more stages to define the location of the lungs.

Response: Although cascade impactor can provide more details on location of drugs; however, Twin Stage Impinger (TSI) is an official instrument to characterize particles and is considered a reliable method for evaluating aerosolization properties. Currently, we don’t have access to cascade impactors and therefore, we have used TSI for evaluating the in-vitro aerosolization properties of the developed formulations.

Please refer to the following papers:

  • https://doi.org/10.1007/s13346-021-01005-5,
  • https://doi.org/10.1016/j.ijpharm.2020.119684,
  • https://doi.org/10.1371/journal.pone.0249683,
  • https://doi.org/10.34172/PS.2023.30,
  • https://doi.org/10.1016/j.jddst.2021.102356,
  • https://doi.org/10.1016/j.jconrel.2024.06.044,
  • https://doi.org/10.1016/j.ijpharm.2022.122492.

Reviewer 3 Report

Comments and Suggestions for Authors

The aim of this study is interesting, but there is not any issues or data to support your conclusion for "against lung cancer".

I think you should added the cell experiments and even in vivo animal experiments to support your purpose of this nanoparticle-system.

Author Response

Q1: The aim of this study is interesting, but there is not any issues or data to support your conclusion for "against lung cancer".

I think you should added the cell experiments and even in vivo animal experiments to support your purpose of this nanoparticle-system.

Response: Thank you for this comment. The cytotoxicity and cell culture experiments are ongoing and will be presented in our future paper. This manuscript mainly focuses on the development and physicochemical characterization of the formulations. However, the term “against lung cancer” has been removed from the text and lung cancer has been mentioned as an example of chronic pulmonary diseases and a possible indication for the developed formulation, as ivermectin has proven to be effective against cancer demonstrated in the published articles (below).

  • https://doi.org/10.1016/j.phrs.2020.105207
  • https://doi.org/10.3389/fphar.2021.717529
  • https://doi.org/10.1007/s00280-020-04041-z

Round 2

Reviewer 1 Report

Comments and Suggestions for Authors

The paper can be accepted as it is.